# Fabrication and evaluation of complicated microstructures on cylindrical surface

**Jiahao Yong**◉, **Junfeng Liu**◉*, **Chaoliang Guan**◉, **Yifan Dai**‡, **Fei Li**‡, **Zhanbin Fan**‡

Laboratory of Science and Technology on Integrated Logistics Support, College of Artificial Intelligence, National University of Defense Technology, Changsha, China

◉ These authors contributed equally to this work.
‡ These authors also contributed equally to this work.
* ljf20090702122@163.com

## Abstract

Various items of roll molds are popularly used to fabricate different kinds of optical films for optoelectronic information and other new and high-tech fields, while the fabrication and evaluation of optical microstructures on a cylindrical roller surface is more difficult than ecumenically manufactured products. In this study, the machinability of microstructures on the roll based on a fast tool servo (FTS) system is investigated. First, the flexible hinge holder for a FTS is designed and its structural parameters are optimized with finite-element analysis and fatigue reliability theory. The tool radius compensation algorithm for complicated microstructures is then deduced based on the surface fitting and bilinear interpolation algorithm of discrete data. Meanwhile, the evaluation index and method are proposed by the medium section method. Finally, a machining test of aspheric arrays on a cylindrical aluminum surface is carried out, and the high quality of the microstructure indicates that the proposed method is able to be used to fabricate optical microstructures.

## Introduction

Optical films with various kinds of microstructures are extensively used in optical instruments, LED lighting, liquid-crystal display, and other applications [1–3]. The microstructure molding of optical films is reliant on the roller [2, 3], the surface of which is graduated in various microstructures. The patterns of optical film microstructures become increasingly complicated with their wide application, while the requirements of shape precision and surface roughness become higher [4]. Therefore, the machining technology of microstructures on roller surfaces is a key research direction in the fabrication of optical films [5].

The mainstream machining technology of microstructures is ultra-precision turning combined with the fast tool servo (FTS) supported by a piezo actuator, which has advantages of high peak acceleration and bandwidth [6, 7]. The United States and Germany are leaders in the relevant fields. For example, the various skills of DRL-type roll lathes of PRECITECH and TDM-series roll lathes of KUGLER are full-blown, and are those roll lathes are highly commercialized. Based on the DRL 1430 roll lathe and FTS 70, workers at Cranfield University (UK) manufactured a series sinusoidal grids with high precision [8]. Workers at the National University of Defense Technology (China) carried out significant work in the fabrication of

**Data Availability Statement:** All relevant data are within the manuscript and its Supporting Information files.

**Funding:** This work was funded by Science Challenge Project of China under grant number

TZ2018006-0202-03/-0102-04 awarded to CG, Science and Technology Foundation of State Key laboratory under grant number No.6142003190101 awarded to JL and Natural National Science Foundation of China (https://isisn.nsfc.gov.cn/egrantweb/) under grant number No.51991372/51991371 awarded to YD. The funders had no role in study design, data collection and analysis, decision to publish, or preparation of the manuscript.

**Competing interests:** The authors have declared that no competing interests exist.

microstructures on the end face with a home-grown FTS system, e.g., utilizing sinusoidal radiation, a sinusoidal phase plane, and a micro-lens array, in which the surface roughness of the lens array with circumferential arrangement reaches 11.4 nm [9].

Lu *et al.* designed a FTS system for precision machining and investigated the influence of the clearance angle of a diamond tool on the microstructure's shape precision [10]. Based on the modification of the tool path, Yu *et al.* put forward a compensation method to pre-compensate the induced profile errors in FTS diamond turning, and effectively improved the machining precision of microstructures [11]. To predict the microstructure generation, Lu *et al.* proposed a method for calculating the microstructure profile amplitude and wavelength, and conducted a cutting test to validate the presented method [12]. Zhu *et al.* studied the feasibility of a novel quasi-elliptical tool servo for vibration suppression in the turning of micro-lens arrays, and effectively decreased the tool vibration caused by the motion non-smoothness in the FTS [13]. With the use of an error-correcting algorithm, Wang *et al.* performed a lens arrays with a 0.06-μm peak-to-valley value using the FTS system [14].

There is no uniform standard for the evaluation of microstructure surfaces at present due their complexity and particularity. The evaluation of surface quality for a single microstructure is generally shape error and surface roughness [15]. Zhou *et al.* developed some methods, such as pattern and characteristic parameter analysis methods, evaluation methods based on inherent characteristics of curved surfaces, and a template matching method, to measure and evaluate microstructure surfaces [16]. A series of evaluation parameters were determined in which the evaluation of microstructure surfaces from single to whole was implemented, aiding the error diagnosis and process optimization [16, 17].

The machining equipment, processing techniques, error compensation methods, and precision evaluation for microstructures were deeply discussed in the above-mentioned studies. While the complexity of the microstructure pattern makes the fabrication and evaluation of the microstructures on a roller surface more difficult, the reliability of the design on FTS must be strengthened. In response, in this study, a tool radius compensation algorithm and an evaluation method for complicated microstructures are proposed, and a machining experiment of aspheric arrays on a cylindrical aluminum surface based on a designed FTS with the consideration of fatigue reliability performed.

## Identification of FTS system

A FTS system is generally driven by a piezoelectric actuator and its motion transferred by a flexible hinge holder; the hinge usually adopts the straight beam structure according to the performance requirements of the system [18]. Fig 1 is a diagram of a straight beam hinge.

Taking the minimal thickness $t$ and beam length $L$ that most affect the behaviors as design factors. With the restrains on the positions of hinge bolt holes and the load on the part in contact with piezoelectric ceramics, the influences of $t$ and $L$ on the hinge stiffness are shown in Fig 2 based on an ANSYS simulation, in which the other structure parameters are in accord with those in Ref. [18]. It can be seen from the figure that the stiffness increases with minimal thickness $t$ and decreases with beam length $L$.

The FTS system makes high-frequency reciprocating motions under working conditions, and the alternating stress on it tends to lead to fatigue failure, which means that the design method for a FTS based on static strength is likely to lead to pre-mature fatigue failure or over-design of the system [19].

Supposing that $f(S)$ and $f(\sigma)$ are the probability density functions of stress $S$ and strength $\sigma$, respectively, the fatigue reliability $\bar{R}$ can be obtained by

$$\bar{R} = P(\sigma > S) = \int_{-\infty}^{\infty} f(S)\left[\int_{S}^{\infty} f(\sigma)\mathrm{d}\sigma\right]\mathrm{d}S \tag{1}$$

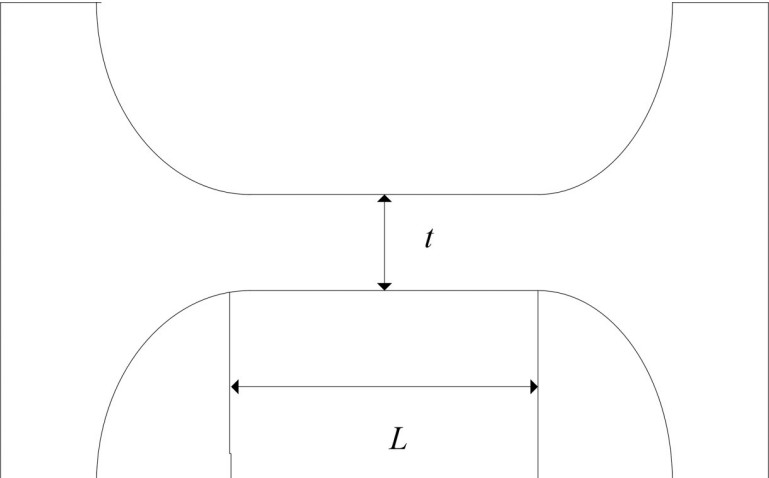

**Fig 1. Diagram of hinge with straight beam structure.**

Where $S$ and $\sigma$ are functions of random variables, such as material property, dimension, and environment variable, and the determinations of their probability density functions must synthesize the distributions of related parameters into comprehensive distributions. Based on the moment method, $S$ can be expressed by

$$S = f(x_1, x_2, x_3, \ldots, x_n) \tag{2}$$

Where $x_i$ ($i = n$) are the design variables.

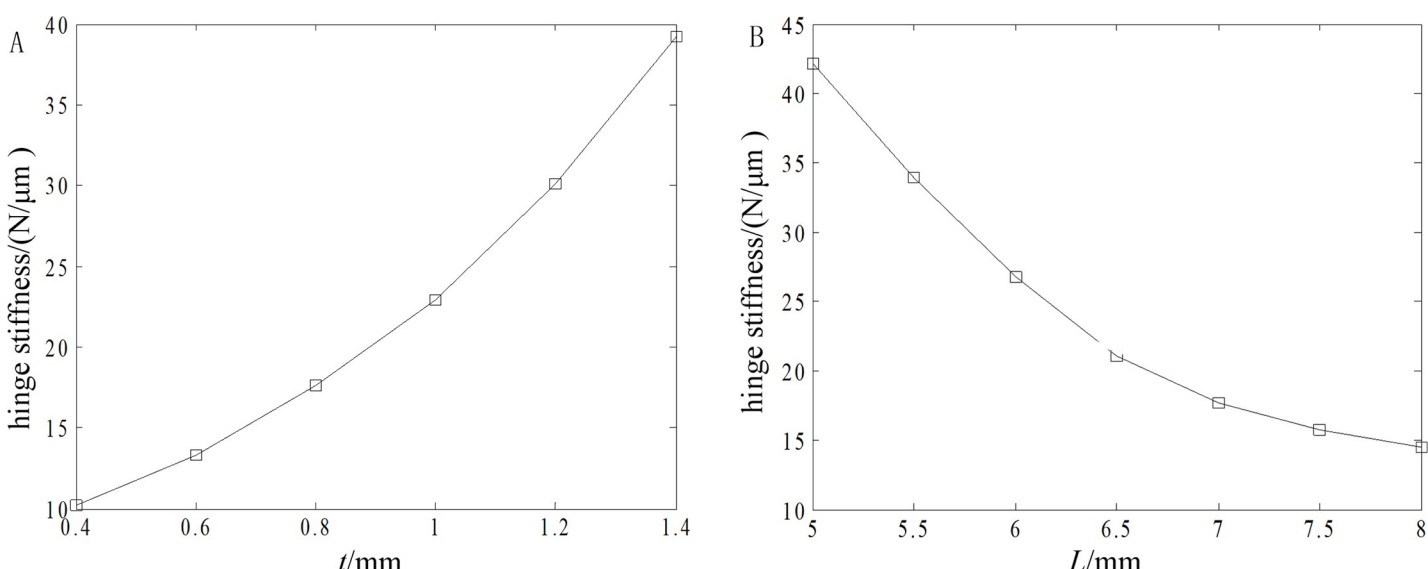

**Fig 2. Influences of minimal thickness t and beam length L on hinge stiffness.** (A) Minimal thickness and (B) Beam length.

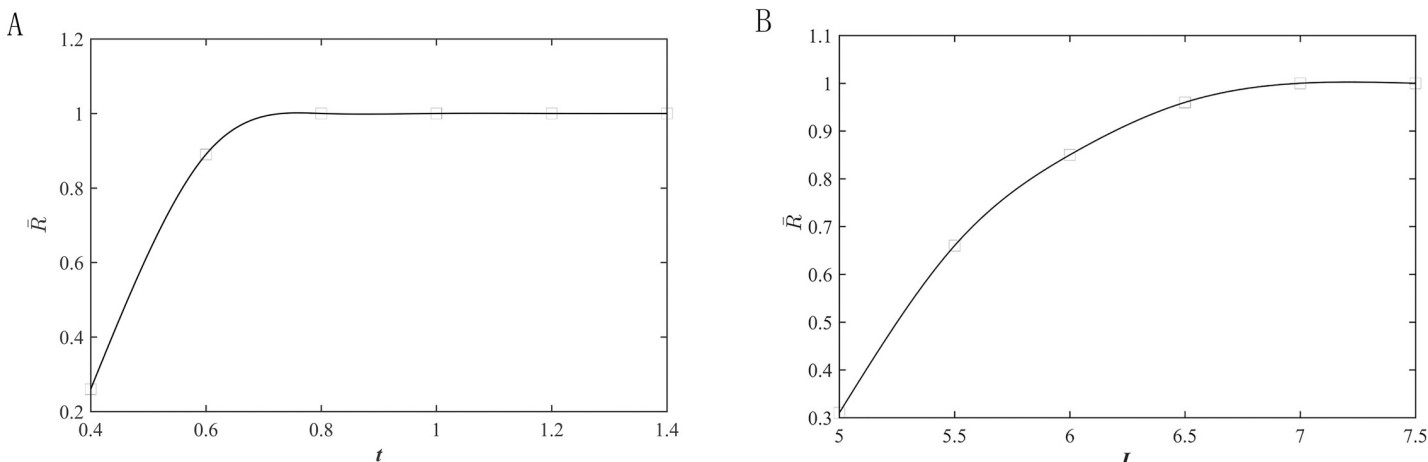

**Fig 3. Influences of minimal thickness t and beam length L on fatigue reliability $\bar{R}$.** (A) Minimal thickness and (B) Beam length.

The moment method supposes that $S$ observes the normal distribution, $S \sim N(u_s, \delta_s^2)$, and the approximations of the mean value $u_s$ and standard deviation $\delta_s$ can be obtained by

$$u_s = E(S) \approx f(x_1, x_2, x_3, \ldots, x_n) + \frac{1}{2}\sum_{i=1}^{n}\frac{\partial^2 f(x)}{\partial x_i^2}\Big|_{x_i=u_i} \cdot \mathrm{var}(x_i)$$

$$\cdot \mathrm{var}(x_i) \qquad (3)$$

$$\delta_s^2 = \mathrm{var}(S) \approx \sum_{i=1}^{n}\left\{\frac{\partial f(x)}{\partial x_i}\Big|_{x_i=u_i}\right\}$$

Where $E(S)$ and $\mathrm{var}(S)$ are the first-order origin moment and second-order central moment about S, respectively.

With the combination of Eqs (1) and (3), the fatigue reliability $\bar{R}$ can be obtained. Fig 3 shows the influences of $t$ and $L$ on $\bar{R}$, and $\bar{R}$ increases with $t$ and decreases with $L$.

According to the results in Figs 2 and 3, the values of $t$ and $L$ are 0.8 and 7 mm, respectively. Fig 4 presents the stress distribution with a 40-μm displacement and the first-order mode shapes of the optimized hinge, and the results are far less than their corresponding critical values. The hinge is obtained by milling and wire cutting.

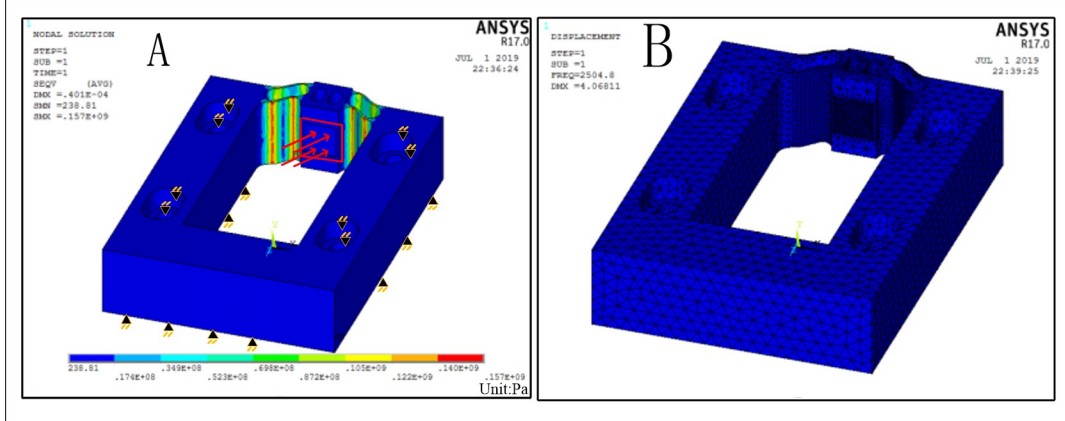

**Fig 4. Simulations of stress distribution and first natural frequency.** (A) Stress distribution and (B) First natural frequency.

## Compensation of tool radius

In the process of ultra-precision machining for arbitrary surfaces, the programming path is the tool center path. The arc radius of the diamond tool tip is $r_\varepsilon$, and the microstructure contour curve is the envelope at a distance $r_\varepsilon$ from the central trajectory of the diamond tool. To obtain the microstructure contour, it is necessary to compensate the tool radius, except for the pyramid microstructure machined by a sharp tool [20].

The microstructure on a cylindrical surface is expressed by $x = g(z, \theta)$, where z and $\theta$ are the height of the microstructure on the cylinder and the angle of the microstructure in the circumferential direction. The central trajectory of the diamond tool after tool radius compensation can be obtained according to Fig 5.

The curve of microstructure surface on a certain section is the section curve and when the tool moves on the section curve, the path of the tool radius is the tool center path. Supposing that $AB$ and $A_1B_1$ are the section curve and tool center path, respectively, and that the spacing between them is $r_\varepsilon$, the slope of any point $A(z_0, x_0)$ on $AB$ at $\theta$ is

$$tg\beta = \frac{\partial x}{\partial z_0} = \frac{\partial f}{\partial z_0} \tag{4}$$

Where $\beta$ is the slope angle.

The related point $A_1(z_1, x_1)$ on $A_1B_1$ can be obtained by

$$\begin{cases} z_1 = z_0 + r_\varepsilon \sin\beta \\ x_1 = x_0 - r_\varepsilon \cos\beta \end{cases} \tag{5}$$

According to Eqs (4) and (5), the equation of the tool center path at $\theta$ after tool radius compensation is

$$x_1 = f\left(z_1 - \frac{r_\varepsilon \cdot \frac{\partial f}{\partial z_0}}{\sqrt{1 + \left(\frac{\partial f}{\partial z_0}\right)^2}}, \theta\right) - \frac{r_\varepsilon}{\sqrt{1 + \left(\frac{\partial f}{\partial z_0}\right)^2}}. \tag{6}$$

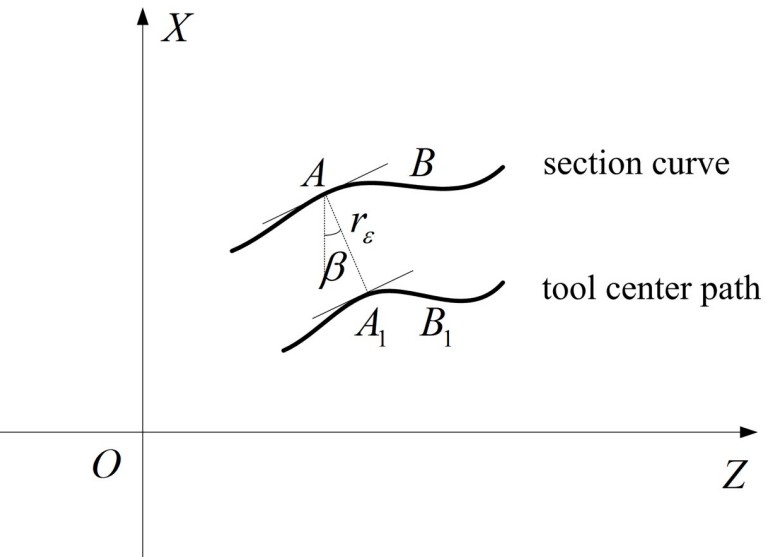

**Fig 5. Section curve and tool center path at angle $\theta$.**

For more complicated microstructure surfaces such as sinusoidal grids and aspherics, the obtainment of the related tool center paths by Eq (6) is complicated. With the calculation for the tool points carried out according to surface discrete points, the related data of the depth of cut can be generated, and the values of the depth of cut near the data points should be obtained by interpolation calculation.

The common mean of interpolation calculation is a nearest-neighbor algorithm, i.e.,

$$g(z, y) = g(i, j), \left( i - \frac{1}{2} \leq z \leq i + \frac{1}{2}, j - \frac{1}{2} \leq y \leq j + \frac{1}{2} \right) \tag{7}$$

Where $g(i, j)$ and $g(z, y)$ are the known data points and interpolation results, respectively.

Although the nearest-neighbor algorithm has a small amount of calculation and fast calculation speed, the calculation precision is not high. The bilinear interpolation algorithm is able to effectively improve the calculation precision, and its core idea is the linear interpolations in two directions.

As shown in Fig 6, $P_{11}$, $P_{21}$, $P_{12}$, and $P_{22}$ are the four known points on the surface, and $P(z, y)$ is any point inside the four points. The linear interpolation in $Z$ direction is

$$f(R_1) = \frac{z_2 - z}{z_2 - z_1} f(P_{11}) + \frac{z - z_1}{z_2 - z_1} f(P_{21})$$
$$f(R_2) = \frac{z_2 - z}{z_2 - z_1} f(P_{12}) + \frac{z - z_1}{z_2 - z_1} f(P_{22}) \tag{8}$$

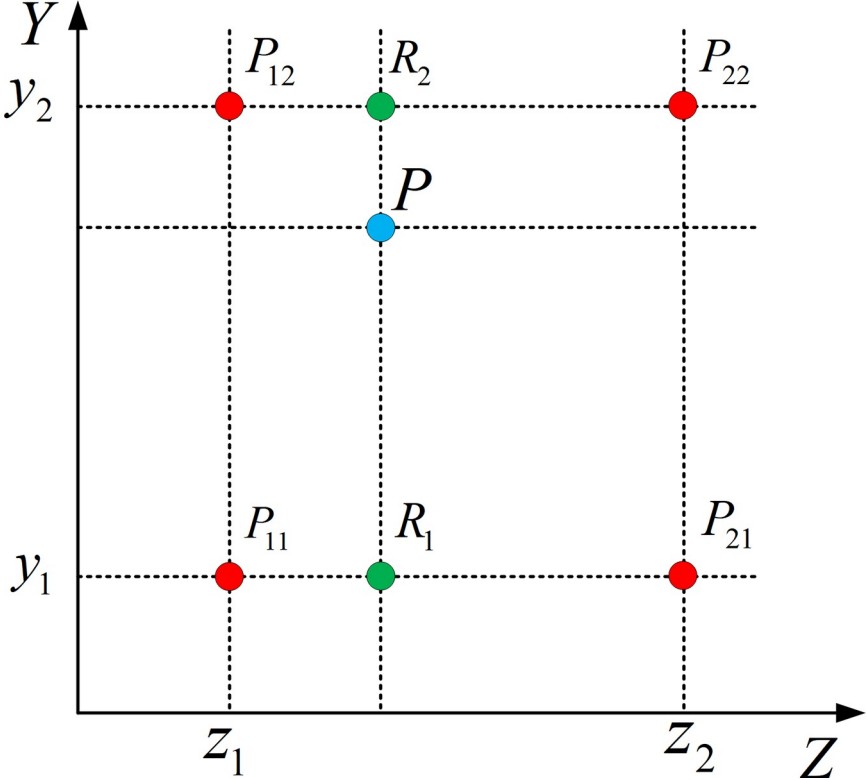

**Fig 6. Schematic diagram of bilinear interpolation algorithm.**

Where $R_1 = (z,y_1)$, $R_2 = (z,y_2)$, $f(P_{11})$, $f(P_{21})$, $f(P_{12})$, $f(P_{22})$, $f(R_1)$, and $f(R_2)$ are the values of the depth of cut of the six points. Next, the linear interpolation in the $Y$ direction is

$$f(P) = \frac{y_2 - y}{y_2 - y_1} f(R_1) + \frac{y - y_1}{y_2 - y_1} f(R_2) \tag{9}$$

Where $f(P)$ is the value of the depth of cut of point $P$.

Based on the least-squares method, the depth-of-cut values of the discrete point are fitted to a surface equation, which can be used for tool radius compensation.

The fabrication of microstructures on cylindrical surface is restricted by the tool geometry, FTS characteristic, and process parameters, which influences the precision and quality of the finished surface. Study of the restriction rule of the manufacturability of such objects is able to determine the selection of tools and process parameters.

## Precision evaluation of microstructure arrays

The machining accuracy is fundamentally determined by the position relationship between workpiece and cutter in the process of machining, and is also closely related to the size of various parameters during processing. For a long time, there has not been a reasonable and unified evaluation standard for cylindrical or roller surface microstructure arrays. Because the array of micro structures is distributed on the cylinder surface, it is very difficult to detect.

The form Talysurf 1240 profilometer is used to measure the maximum profile of the microstructure along the axis of the workpiece, and the surface quality of the microstructure is indirectly evaluated from the roughness and profile deviation of the contour line. Roughness is the high frequency component of contour, which is obtained by high pass filtering method, and the root mean square deviation (RMS) is used to calculate the roughness.

The root-mean-square deviation $S_q$ and contour deviation $S_T$ of the isocline are adopted to evaluate the precision of microstructures on cylindrical surface. $S_q$ and $S_T$ are high- and low-frequency components, respectively, and their formulas are

$$S_q = \sqrt{\frac{1}{N} \sum_{i=1}^{N} x_h(z_i)^2}$$
$$\begin{cases} e_i = x_l(z_i) - x(z_i) \\ S_T = \max(e_i) - \min(e_i) \end{cases} \tag{10}$$

Where $N$ is the number of sampling points; $x_h(z_i)$ and $x_l(z_i)$ are the contour data points after high- and low-pass filtering, respectively; $x(z_i)$ are ideal contour data; $i = 1,2,\ldots,N$.

The ideal and measured contour curves should be matched because their coordinate systems are different. As shown in Fig 7, $XOZ$ and $X'O'Z'$ are the coordinate systems of ideal and measured contours, respectively; $n$ and $n'$ are the normal vectors of ideal and measured contours, respectively; $A,C$ and $A',C'$ are the endpoints of the ideal and measured contours; B' are the lowest points of ideal and measured contours, respectively; B and B' are the lowest points of ideal and measured contours, respectively.

Anti-tilting of the measured contour curves is used to make the normal vectors of the two curves in the same direction. The spacing between the endpoints and lowest point of each contour based on the three-point fix method is regarded as the objective function, and the translation vector $\mathbf{M}$ is obtained by minimum calculation. The measured contour can be effectively

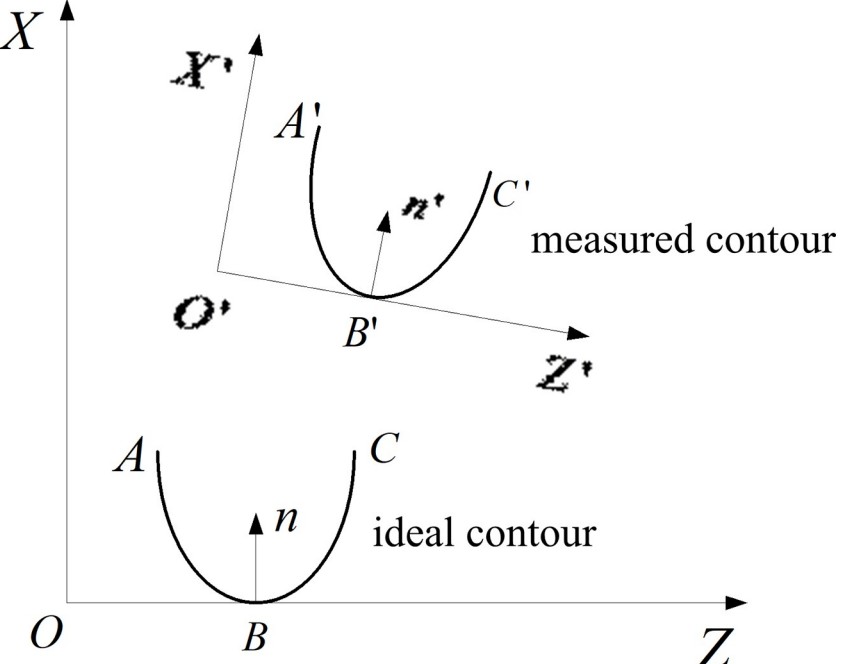

**Fig 7. Coordinate systems of ideal and measured contours.**

and quickly matched with the ideal contour by the following equation:

$$G = \sqrt{(x_A - x_{A'})^2 + (z_A - z_{A'})^2} + \sqrt{(x_B - x_{B'})^2 + (z_B - z_{B'})^2} + \sqrt{(x_C - x_{C'})^2 + (z_C - z_{C'})^2} \\ - |\mathbf{M}| \tag{11}$$

The overall evaluation for microstructure arrays can be determined by the overall depth error $e_h$ and pitch error $e_p$. Their calculation formulas are

$$e_h = \frac{1}{K} \sum_{k=1}^{K} |h_k^{'} - h_k|$$

$$e_p = \frac{1}{K} \sum_{k=1}^{K} |P_k^{'} - P_k| \tag{12}$$

Where $h_k$ and $P_k$ are the ideal $k$th maximal depth of cut and pitch along the axial direction of microstructure arrays, respectively; $h_k$' and $P_k$' are the actual $k$th maximal depth of cut and pitch along the axial direction of microstructure arrays, respectively; $k = 1,2,...,K$. It is worthy of note that the pitch $P_k$ is the spacing between any line parallel to the axial direction of microstructure arrays and the intersection at the same position on the measured contour curve.

## Processing and evaluation of microstructure arrays of aspheric arrays

Owing to their high optical quality, aspheric surface optical parts are widely used in modern photoelectric products. The general mathematical formula for aspherics is

$$x = \frac{cr^2}{1 + \sqrt{1 - (1 + \gamma)c^2 r^2}} + \sum_{i=1}^{N} F_i r^{2(i+1)} \tag{13}$$

**Table 1. Processing parameters.**

| Spindle speed (r·min$^{-1}$) | Cut of depth of optical surface (μm) | Feed speed for optical surface (mm·min$^{-1}$) | Feed speed for arrays (mm·min$^{-1}$) | Length of arrays (mm) | Depth of aspherics (μm) | Arc radius of diamond tool tip $r_\varepsilon$ (mm) |
|---|---|---|---|---|---|---|
| 15 | 10 | 0.4 | 0.4 | 8 | 15.749 | 1 |

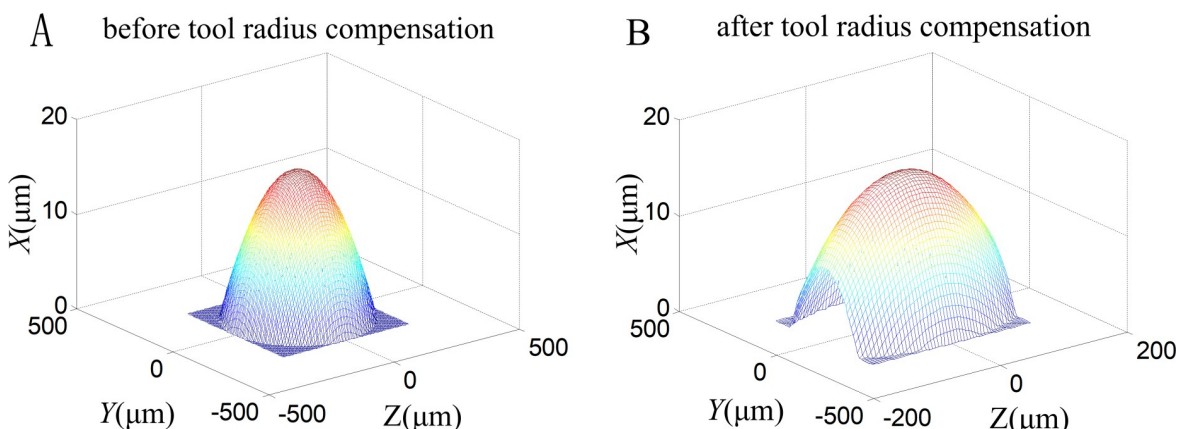

**Fig 8. Simulation images of aspheric surface before and after tool radius compensation.** (A) Before and (B) After tool radius compensation.

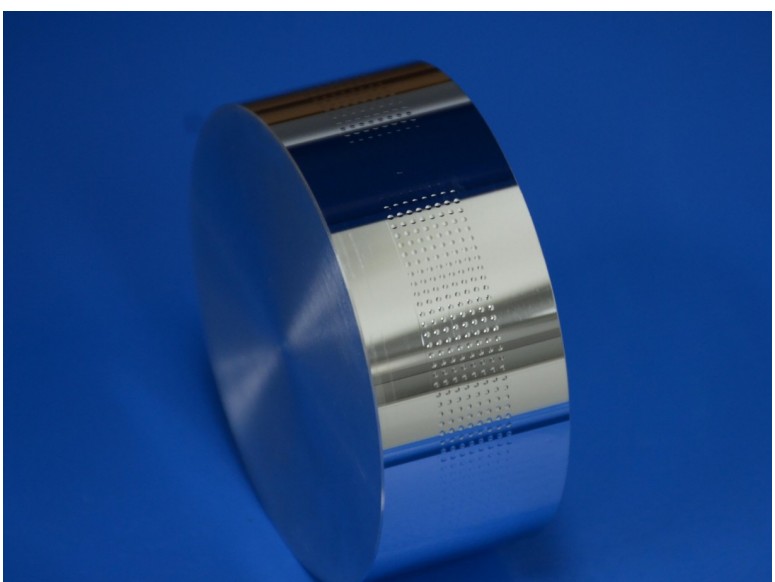

**Fig 9. Processing result of aspheric microstructure arrays on cylindrical surface.**

Where $c = 1/R$, and $R$ is the base circle radius of curve; $\gamma$ is the conic coefficient; $F_i$ are aspheric coefficients.

Based on the NF350-type ultra-precision lathe and designed FTS system, a processing test of aspheric microstructure arrays on a cylindrical aluminum workpiece with a diameter of 50

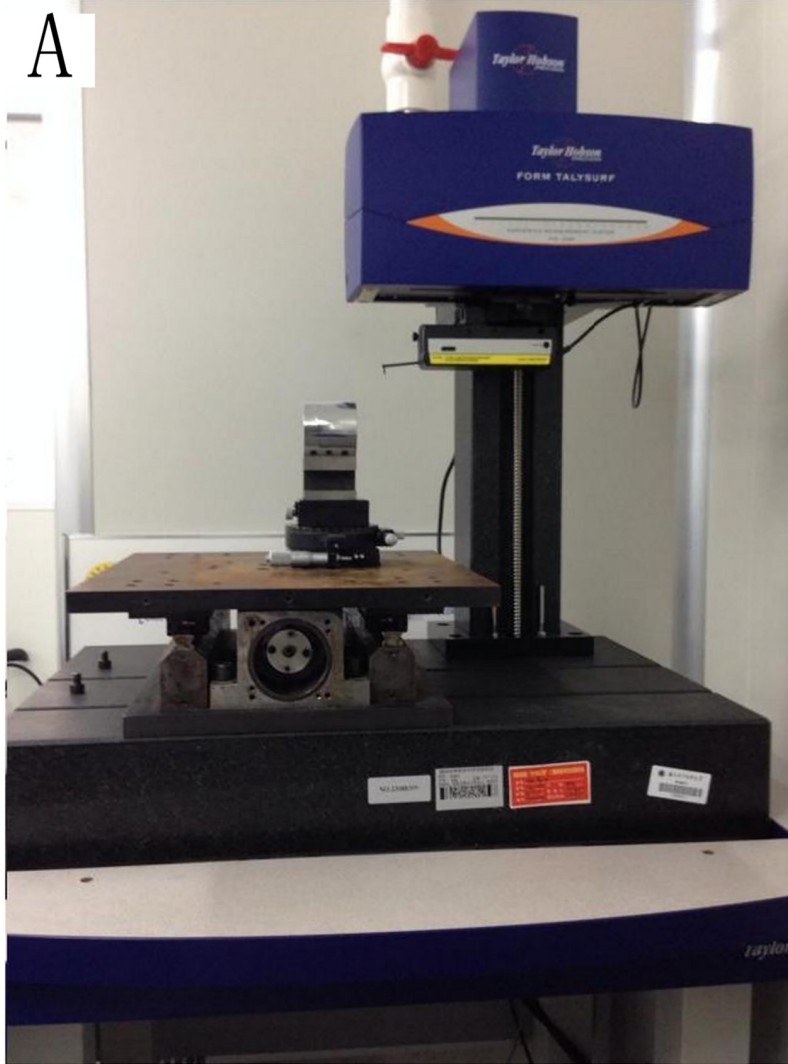

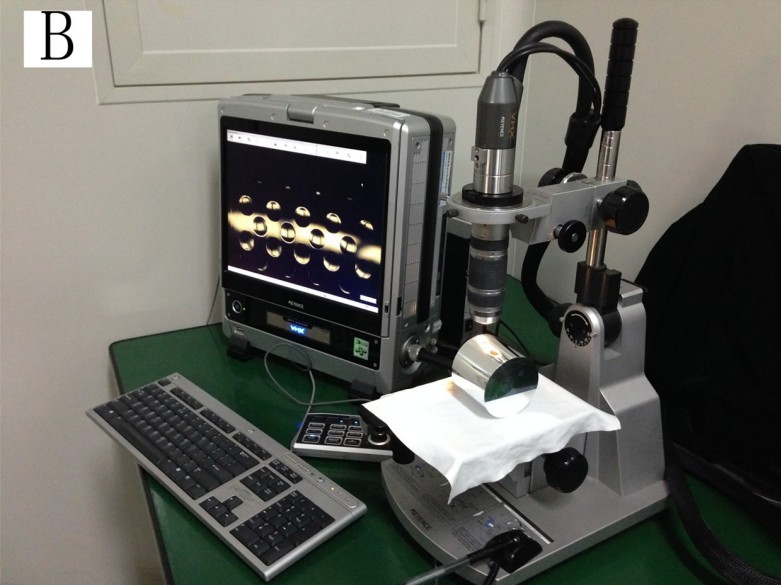

**Fig 10. Measurement scene.** (A) Profilometer and (B) Microscope.

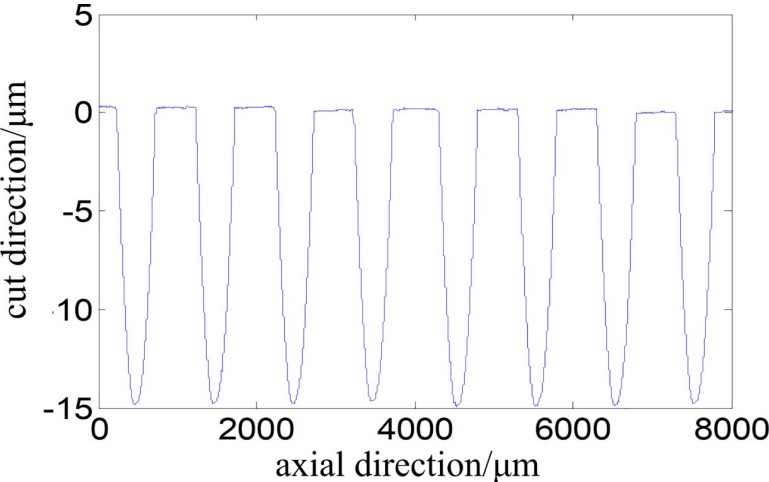

**Fig 11. Contours of studied aspheric arrays.**

mm and height of 20 mm was carried out. The values of $c$, $\gamma$, and $F_i$ were set as 1, 0, and 1/2000, respectively, and other parameters are as shown in Table 1.

According to Eq (6), the tool radius compensation for aspherics is difficult. The aspheric surface is discretized and the tool center position related to each discrete point is calculated, and then the data are converted into grids based on a bilinear interpolation algorithm. Fig 8 shows the simulation images of the aspheric surface before and after tool radius compensation.

The aspheric microstructure arrays on cylindrical surface are processed successfully with the techniques and equipment described above, and the processing result is shown in Fig 9.

Owing to the large height difference and the difficulty of measurement along a circumferential direction, it is difficult for white-light and wavefront interferometers to accurately measure the surface quality of aspheric microstructures on cylindrical surfaces. A Talysurf PGI 1240-type profilometer and VH-Z5002-type microscope were used to examine contour deviation $S_T$ and root-mean-square deviation $S_q$, respectively. Fig 10 shows the measurement scene.

The contours of the aspheric arrays along the axial direction are shown in Fig 11. After measured data analysis, it is known that the values of overall depth error $e_h$ and pitch error $e_p$ are 1.034 and 1.943μm, respectively, and the values of root-mean-square deviation $S_q$ and contour

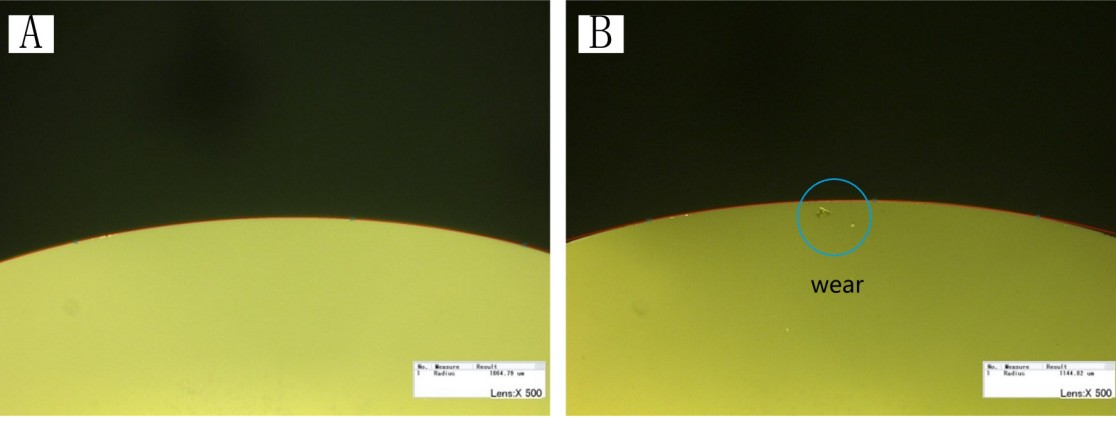

**Fig 12.** Microscopic observations of diamond tool (A) before (**$r_\varepsilon$ = 1064.79 μm**) and (B) after processing (**$r_\varepsilon$ = 1144.82 μm**).

deviation $S_T$ are 0.022 and 0.51 μm, respectively. The results indicate that the proposed FTS system and method are able to effectively process and evaluate the complicated microstructures on cylindrical surfaces.

Fig 12 shows the microscopic observations of the diamond tool before and after processing. There is no damage on the circular tool, while the arc radius $r_\varepsilon$ increases and the increment is 80.03 μm, which indicates that the tool is slightly worn during the working process. The machining of microstructures with the FTS system is a high-frequency motion, and a high-quality tool is necessary.

## Conclusions

The process and evaluation of complicated microstructures on cylindrical surface are investigated, and verification tests carried out based on the designed FTS system and an ultra-precision lathe. The following conclusions are drawn from the results.

a. With the consideration of fatigue reliability, the structure parameters of the straight beam hinge is optimized by ANSYS simulation to obtain high performance of the FTS system.

b. The tool radius compensation for the machining of complicated microstructures is performed based on data discretization and a bilinear interpolation algorithm.

c. Root-mean-square deviation $S_q$, contour deviation $S_T$, overall depth error $e_h$, and pitch error $e_p$ are proposed as evaluation indexes.

d. The process of fabricating aspheric microstructure arrays on cylindrical surfaces indicates the feasibility and reliability of the proposed FTS system and method, which is useful for the fabrication of optical films.

## Supporting information

**S1 File. Compensation algorithm.**
(DOC)

## Acknowledgments

We thank LetPub (www.letpub.com) for its linguistic assistance during the preparation of this manuscript.

## Author Contributions

**Conceptualization:** Jiahao Yong, Junfeng Liu, Chaoliang Guan, Fei Li, Zhanbin Fan.

**Data curation:** Jiahao Yong, Junfeng Liu, Chaoliang Guan, Yifan Dai, Fei Li, Zhanbin Fan.

**Formal analysis:** Jiahao Yong, Junfeng Liu.

**Funding acquisition:** Chaoliang Guan, Yifan Dai.

**Investigation:** Jiahao Yong, Junfeng Liu, Zhanbin Fan.

**Methodology:** Jiahao Yong, Junfeng Liu, Chaoliang Guan, Yifan Dai.

**Project administration:** Jiahao Yong, Junfeng Liu, Chaoliang Guan, Yifan Dai.

**Resources:** Jiahao Yong, Chaoliang Guan, Yifan Dai.

**Software:** Jiahao Yong, Chaoliang Guan, Zhanbin Fan.

**Supervision:** Zhanbin Fan.

**Validation:** Jiahao Yong, Junfeng Liu, Chaoliang Guan, Fei Li.

**Visualization:** Jiahao Yong, Junfeng Liu, Fei Li.

**Writing – original draft:** Jiahao Yong.

**Writing – review & editing:** Jiahao Yong, Junfeng Liu, Chaoliang Guan, Yifan Dai, Fei Li, Zhanbin Fan.

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
