## [Decision Letter · Decision Letter 0]

9 Oct 2020

PONE-D-20-24573

Fabrication and evaluation of complicated microstructures on cylindrical surface

PLOS ONE

Dear Dr. Liu,

Thank you for submitting your manuscript to PLOS ONE. After careful consideration, we feel that it has merit but does not fully meet PLOS ONE’s publication criteria as it currently stands. Therefore, we invite you to submit a revised version of the manuscript that addresses the points raised during the review process.

As detailed in the attached comments, reviewers both agree that the manuscript is worth publishing, but it should undergo a major revision beforehand. In particular, the global organization of the paper and the clarity of both the text and illustrations must be improved. Also clarifications on equations must be provided (see comments from Reviewer #1).

We look forward to receiving your revised manuscript.

Kind regards,

Marco Livesu

Academic Editor

PLOS ONE

Journal Requirements:

2.We suggest you thoroughly copyedit your manuscript for language usage, spelling, and grammar. If you do not know anyone who can help you do this, you may wish to consider employing a professional scientific editing service.  

4. We note you have included a table to which you do not refer in the text of your manuscript. Please ensure that you refer to Table 1 in your text; if accepted, production will need this reference to link the reader to the Table.

<h1>** **</h1>

Reviewers' comments:

Reviewer's Responses to Questions

**Comments to the Author**

1. Is the manuscript technically sound, and do the data support the conclusions?

Reviewer #1: Partly

Reviewer #2: Partly

2. Has the statistical analysis been performed appropriately and rigorously? 

Reviewer #1: N/A

Reviewer #2: No

3. Have the authors made all data underlying the findings in their manuscript fully available?

Reviewer #1: No

Reviewer #2: Yes

4. Is the manuscript presented in an intelligible fashion and written in standard English?

Reviewer #1: No

Reviewer #2: Yes

5. Review Comments to the Author

Reviewer #1: 1. Is the manuscript technically sound, and do the data support the conclusions?

In general, the various technical parts that make up the method seem sound (compensation of tools radius, optimization of the hinge parameters). Most of the methods used to solve them are standard practice (e.g. bilinear interpolation).

However, certain aspects are not clear:

- Could you clarify if the derivation of $\\sigma$ is analogous the $S$ (Equations 2 and 3), such that both quantities are comparable?

- Function $f$ in Equation 2 is the same as function $f$ in Equation 1? Shouldn't it be the inverse function since one reads $f(S) and the other $S=f(x1,x2,...xn)$?

- Equation 3: how are the second and first order differentials calculated?

- Line 122: another different function named $f$, I recommend to change the nomenclature. Could you also precise what $z$ and $\\theta$ are in this function?

- Line 127: could you define beforehand what is "section curve" and "tool center path"?

- Equation 7: another function named $f$ which I assume is different from the previous?

The precision of the method is backed up with comparisons between the ground truth and the result considering some error metrics (section "Precision evaluation of microstructure arrays"). The results seem to indicate that the procedure decreases the error of the fabrication process.

2. Has the statistical analysis been performed appropriately and rigorously?

N/A.

3. Have the authors made all data underlying the findings in their manuscript fully available?

I was unable to find the data related to the different plots. Also, It would be appreciated if the authors made the code of the "compensation algorithm" available as well.

4. Is the manuscript presented in an intelligible fashion and written in standard English?

The descriptive and technical content of the manuscript is often difficult to follow, mostly due to unusual phrasing and grammar.

Authors should seek help in correcting and polishing the text, otherwise, it becomes sometimes an uphill task to read it.

Besides, it would be appreciated it the main goal of the manuscript is described in more detail in the introduction section, along with the corresponding challenges.

Reviewer #2: This paper presents a pipeline to optimize the machinability of the FTS system used to fabricate microstructures on roll molds. The authors first optimize the reliability of the fabrication system by selecting a proper design parameter of a flexible hinge holder. A tool radius compensation method is used to enhance the fabrication precision, where the least-square solution is used to approximate the cut depth. To verify the fabrication result, an evaluation method is generated to reduce the evaluation error.

I believe the contribution of this work is solid as the precision of the final fabricated microstructure is greatly enhanced and seems aspheric surface is successfully fabricated. Although it needs to be mention that the tool radius compensation method used in the paper is not novel in the precision fabrication field. The manuscript is also not well-organized, and I have the sense that each part is weekly connected and not properly justified. My main concern is that most figures are not well prepared that cannot give enough information to support their conclusions. Detailed comments are as follows.

* Line 80, please add the description of where the hinge holder is used in the FTS system (better to add a figure to illustrate the fabrication process). And please make it clear how the FEM simulation is applied to the component and describe how the boundary condition and workload are set.

* The resolution of Fig 4 is very low, and the data is the loss of the unit.

* Line 137, why the aspheric is hard to be obtained by Eq.6? Since designed aspheric surfaces can always have an implicit function to explain the surface and used in parameterization function.

* The section “precision evaluation of microstructure arrays” is hard to follow. Please first add the explanation where the evaluation error comes from.

* The captain of Fig 6 cannot match with the figure itself.

* Fig 8, the target aspheric structure should be added as a ground-truth for reference.

* Fig.12 basically gives no useful information as the ground-truth of the surface is not shown. Please also add a reference distance scale in the figure.

In summary, I do think this paper requires heavily modify before it is ready for publication. The author should make the paper well organized and make all the figures easy to understand and technical details clear to make their claim self-contained.

6. PLOS authors have the option to publish the peer review history of their article (what does this mean?). If published, this will include your full peer review and any attached files.

Reviewer #1: No

Reviewer #2: No

---

## [Author Response · Author response to Decision Letter 0]

16 Oct 2020

Journal Requirements:

When submitting your revision, we need you to address these additional requirements

2.We suggest you thoroughly copyedit your manuscript for language usage, spelling, and grammar. If you do not know anyone who can help you do this, you may wish to consider employing a professional scientific editing service. 

Response: We regret there were problems with the English. The paper has been carefully revised by a professional language editing service from LetPub to improve the grammar and readability. LetPub is an author service brand owned and operated by Accdon LLC. Headquartered in the Boston area, It is a full-spectrum author services company with a large team of US-based certified language and scientific editors, ISO 17001 accredited translators, and professional scientific illustrators and animators.

 Response:The ORCID iD of the corresponding author is 0000-0002-7089-2388. I have updated my information in Editorial Manager.

4. We note you have included a table to which you do not refer in the text of your manuscript. Please ensure that you refer to Table 1 in your text; if accepted, production will need this reference to link the reader to the Table.

Response: Table 1 referred in this paper was written as Figure 1, which has been corrected.

Reviewers' comments:

Reviewer's Responses to Questions

Comments to the Author

1. Is the manuscript technically sound, and do the data support the conclusions?

Reviewer #1: Partly

Reviewer #2: Partly

2. Has the statistical analysis been performed appropriately and rigorously?

Reviewer #1: N/A

Reviewer #2: No

3. Have the authors made all data underlying the findings in their manuscript fully available?

Reviewer #1: No

Reviewer #2: Yes

4. Is the manuscript presented in an intelligible fashion and written in standard English?

Reviewer #1: No

Reviewer #2: Yes

5. Review Comments to the Author

Reviewer #1: 1. Is the manuscript technically sound, and do the data support the conclusions?

In general, the various technical parts that make up the method seem sound (compensation of tools radius, optimization of the hinge parameters). Most of the methods used to solve them are standard practice (e.g. bilinear interpolation).

However, certain aspects are not clear:

- Could you clarify if the derivation of $\\sigma$ is analogous the $S$ (Equations 2 and 3), such that both quantities are comparable?

Response: Response: Stress S andσ strength are functions of many random variables, such as material properties, size parameters and environmental variables. The degree function is very difficult to obtain directly. It is necessary to synthesize the distribution of relevant parameters into the distribution of stress and strength Including algebraic method, moment method and Monte Carlo method

- Function $f$ in Equation 2 is the same as function $f$ in Equation 1? Shouldn't it be the inverse function since one reads $f(S) and the other $S=f(x1,x2,...xn)$?

Response: They are not a function, the probability density function of stress in equation (1) and the variable function of stress based on moment method in equation (2).

- Equation 3: how are the second and first order differentials calculated?

Response: It can be obtained by algebraic method or moment method.

- Line 122: another different function named $f$, I recommend to change the nomenclature. Could you also precise what $z$ and $\\theta$ are in this function?

Response: The terminology and supplementary instructions have been changed as required by the reviewer.

- Line 127: could you define beforehand what is "section curve" and "tool center path"?

Response: Pre description has been made in accordance with reviewer opinion.

- Equation 7: another function named $f$ which I assume is different from the previous?

Response:They are different, and we changed the function symbol.

The precision of the method is backed up with comparisons between the ground truth and the result considering some error metrics (section "Precision evaluation of microstructure arrays"). The results seem to indicate that the procedure decreases the error of the fabrication process.

Response: In this chapter, the simulation results are compared, and then the evaluation method is applied to the later processing evaluation.

2. Has the statistical analysis been performed appropriately and rigorously?

N/A.

3. Have the authors made all data underlying the findings in their manuscript fully available?

I was unable to find the data related to the different plots. Also, It would be appreciated if the authors made the code of the "compensation algorithm" available as well.

Response: We will upload the compensation algorithm programming steps as supplementary materials. The filename is S1_Compensation Algorithm.

4. Is the manuscript presented in an intelligible fashion and written in standard English?

The descriptive and technical content of the manuscript is often difficult to follow, mostly due to unusual phrasing and grammar.

Authors should seek help in correcting and polishing the text, otherwise, it becomes sometimes an uphill task to read it.

Besides, it would be appreciated it the main goal of the manuscript is described in more detail in the introduction section, along with the corresponding challenges.

Response:It has been revised according to the reviewer' opinions. We regret there were problems with the English. The paper has been carefully revised by a professional language editing service from LetPub to improve the grammar and readability. LetPub is an author service brand owned and operated by Accdon LLC. Headquartered in the Boston area, It is a full-spectrum author services company with a large team of US-based certified language and scientific editors, ISO 17001 accredited translators, and professional scientific illustrators and animators.

Reviewer #2: This paper presents a pipeline to optimize the machinability of the FTS system used to fabricate microstructures on roll molds. The authors first optimize the reliability of the fabrication system by selecting a proper design parameter of a flexible hinge holder. A tool radius compensation method is used to enhance the fabrication precision, where the least-square solution is used to approximate the cut depth. To verify the fabrication result, an evaluation method is generated to reduce the evaluation error.

I believe the contribution of this work is solid as the precision of the final fabricated microstructure is greatly enhanced and seems aspheric surface is successfully fabricated. Although it needs to be mention that the tool radius compensation method used in the paper is not novel in the precision fabrication field. The manuscript is also not well-organized, and I have the sense that each part is weekly connected and not properly justified. My main concern is that most figures are not well prepared that cannot give enough information to support their conclusions. Detailed comments are as follows.

* Line 80, please add the description of where the hinge holder is used in the FTS system (better to add a figure to illustrate the fabrication process). And please make it clear how the FEM simulation is applied to the component and describe how the boundary condition and workload are set.

Response: Relevant supplementary explanations have been made according to the advice.

* The resolution of Fig 4 is very low, and the data is the loss of the unit.

Response: Figure 4 is the cloud image saved by ANSYS software, which is already the highest resolution.

* Line 137, why the aspheric is hard to be obtained by Eq.6? Since designed aspheric surfaces can always have an implicit function to explain the surface and used in parameterization function.

Response: The expression in the text is not accurate, and the meaning of the expression has been revised according to the suggestion

* The section “precision evaluation of microstructure arrays” is hard to follow. Please first add the explanation where the evaluation error comes from.

Response: Relevant supplementary explanations have been made according to the advice.

* The captain of Fig 6 cannot match with the figure itself.

Response: The captain of Fig 6 is corrected according to the advice.

* Fig 8, the target aspheric structure should be added as a ground-truth for reference.

Response: Figure 8 shows the simulation diagram before and after the tool radius compensation. If you refer to figure 9, the first picture has been taken on the blue tape desktop to facilitate the observation of microstructure

* Fig.12 basically gives no useful information as the ground-truth of the surface is not shown. Please also add a reference distance scale in the figure.

Response: Figure 12 shows wear, which has been remarked.

In summary, I do think this paper requires heavily modify before it is ready for publication. The author should make the paper well organized and make all the figures easy to understand and technical details clear to make their claim self-contained.

6. PLOS authors have the option to publish the peer review history of their article (what does this mean?). If published, this will include your full peer review and any attached files.

Do you want your identity to be public for this peer review? For information about this choice, including consent withdrawal, please see our Privacy Policy.

Reviewer #1: No

Reviewer #2: No

---

## [Decision Letter · Decision Letter 1]

3 Nov 2020

PONE-D-20-24573R1

Fabrication and evaluation of complicated microstructures on cylindrical surface

PLOS ONE

Dear Dr. Liu,

Thank you for submitting your manuscript to PLOS ONE. After careful consideration, we feel that it has merit but does not fully meet PLOS ONE’s publication criteria as it currently stands. Therefore, we invite you to submit a revised version of the manuscript that addresses the points raised during the review process.

Specifically, Figure 4 still lacks the unit of the data and the application points of the boundary conditions are not highlighted (see comments from Reviewer 2).

We look forward to receiving your revised manuscript.

Kind regards,

Marco Livesu

Academic Editor

PLOS ONE

Reviewers' comments:

Reviewer's Responses to Questions

**Comments to the Author**

1. If the authors have adequately addressed your comments raised in a previous round of review and you feel that this manuscript is now acceptable for publication, you may indicate that here to bypass the “Comments to the Author” section, enter your conflict of interest statement in the “Confidential to Editor” section, and submit your "Accept" recommendation.

Reviewer #1: All comments have been addressed

Reviewer #2: (No Response)

2. Is the manuscript technically sound, and do the data support the conclusions?

Reviewer #1: Yes

Reviewer #2: Yes

3. Has the statistical analysis been performed appropriately and rigorously? 

Reviewer #1: N/A

Reviewer #2: N/A

4. Have the authors made all data underlying the findings in their manuscript fully available?

Reviewer #1: Yes

Reviewer #2: Yes

5. Is the manuscript presented in an intelligible fashion and written in standard English?

Reviewer #1: Yes

Reviewer #2: Yes

6. Review Comments to the Author

Reviewer #1: Thank you for attempting to answer my questions and make the corresponding changes.

One minor comment, when referring to the "nearest-neighbor algorithm" (e.g. in line 153), be aware that this is different (compared to the bilinear interpolation method used in the paper, i.e. has a lower degree (https://en.wikipedia.org/wiki/Nearest-neighbor_interpolation).

Reviewer #2: The authors have made substantial changes to the manuscript in this review round and addressed most of my comments. I appreciate the effort of adding the supplemental document to describe the detail of their program.

There's one comment missed. Fig 4(a) still lacks the unit of the data and the author didn't show where they set the boundary condition. In which area the holder is contact with piezoelectric ceramics? It’s recommended to highlight the contact region with some symbol, add the unit of the data, and remove the file path at the bottom of this figure. Meanwhile, please rephrase the caption of Fig 4(b).

7. PLOS authors have the option to publish the peer review history of their article (what does this mean?). If published, this will include your full peer review and any attached files.

Reviewer #1: No

Reviewer #2: No

---

## [Author Response · Author response to Decision Letter 1]

10 Nov 2020

Respond to the academic editor Marco Livesu:

Thank you for your careful review. I have revised figure 4 according to reviewer 2's comments.

Respond to the comments by Reviewer #1

Reviewer #1: Thank you for attempting to answer my questions and make the corresponding changes.

One minor comment, when referring to the "nearest-neighbor algorithm" (e.g. in line 153), be aware that this is different (compared to the bilinear interpolation method used in the paper, i.e. has a lower degree (https://en.wikipedia.org/wiki/Nearest-neighbor_interpolation).

Response：

Thanks for your exhaustive advice. I'm sorry that I didn't make this part clear. As you said, the "nearest-neighbor algorithm" is different from to the bilinear interpolation method, And the former has a lower degree. The purpose of the nearest neighbor algorithm mentioned in this paper is to reflect the advantages of the bilinear interpolation method in calculation accuracy. We have reorganized this section in the paper.

Respond to the comments by Reviewer #2

Reviewer #2: The authors have made substantial changes to the manuscript in this review round and addressed most of my comments. I appreciate the effort of adding the supplemental document to describe the detail of their program.

There's one comment missed. Fig 4(a) still lacks the unit of the data and the author didn't show where they set the boundary condition. In which area the holder is contact with piezoelectric ceramics? It’s recommended to highlight the contact region with some symbol, add the unit of the data, and remove the file path at the bottom of this figure. Meanwhile, please rephrase the caption of Fig 4(b).

Response：

Thank you for your reminding. I'm sorry for the missing comment. The data unit of the Fig 4(a) has been added. We highlighted the location of boundary conditions with black triangle mark, the contact region with red diamond box, and the applied load with red arrow the in Fig 4(a). The file path at the bottom of this figure has also been removed. 

The unit of stress in Fig 4(a) is Pascal (Pa). The bottom surface of the hinge and the top and side of the bolt counterbore are fixed according to the actual working conditions. The load is applied on the contact surface, which is highlighted by a red rectangle, between the piezoelectric ceramic front face and the moving block of the hinge.

The caption of Fig 4(b) has been renamed with first natural frequency.

---

## [Editor Report · Decision Letter 2]

12 Nov 2020

Fabrication and evaluation of complicated microstructures on cylindrical surface

PONE-D-20-24573R2

Dear Dr. Liu,

We’re pleased to inform you that your manuscript has been judged scientifically suitable for publication and will be formally accepted for publication once it meets all outstanding technical requirements.

Kind regards,

Marco Livesu

Academic Editor

PLOS ONE
---

## [Editor Report · Acceptance letter]

1 Dec 2020

PONE-D-20-24573R2 

Fabrication and evaluation of complicated microstructures on cylindrical surface 

Dear Dr. Liu:

I'm pleased to inform you that your manuscript has been deemed suitable for publication in PLOS ONE. Congratulations! Your manuscript is now with our production department. 

Kind regards, 

on behalf of

Dr. Marco Livesu 

Academic Editor

PLOS ONE